# New Insights into the Role of the *Trypanosoma cruzi* Aldo-Keto Reductase *Tc*AKR

**DOI:** 10.3390/pathogens12010085

**Published:** 2023-01-05

**Authors:** Florencia Díaz-Viraqué, María Laura Chiribao, Lisvane Paes-Vieira, Matias R. Machado, Paula Faral-Tello, Ramiro Tomasina, Andrea Trochine, Carlos Robello

**Affiliations:** 1Laboratorio de Interacciones Hospedero Patógeno, Unidad de Biología Molecular, Institut Pasteur de Montevideo, Montevideo 11400, Uruguay; 2Departamento de Bioquímica, Facultad de Medicina Universidad de la República, Montevideo 11400, Uruguay; 3Unidad de Proteínas Recombinantes, Institut Pasteur de Montevideo, Montevideo 11300, Uruguay; 4Laboratory of Apicomplexan Biology, Institut Pasteur de Montevideo and Departamento de Parasitología, Facultad de Medicina Universidad de la República, Montevideo 11300, Uruguay; 5Centro de Referencia en Levaduras y Tecnología Cervecera (CRELTEC), Instituto Andino Patagónico de Tecnologías Biológicas y Geoambientales (IPATEC), CONICET-Universidad Nacional del Comahue, Quintral 1250, San Carlos de Bariloche 8400, Argentina

**Keywords:** *Trypanosoma cruzi*, aldo-keto reductase, mitochondrial enzyme, kinetoplast, antipodal sites, prostaglandin F_2_α synthase, nifurtimox metabolism

## Abstract

Chagas disease is a zoonotic infectious disease caused by the protozoan parasite *Trypanosoma cruzi*. It is distributed worldwide, affecting around 7 million people; there is no effective treatment, and it constitutes a leading cause of disability and premature death in the Americas. Only two drugs are currently approved for the treatment, Benznidazole and Nifurtimox, and both have to be activated by reducing the nitro-group. The *T. cruzi* aldo-keto reductase (*Tc*AKR) has been related to the metabolism of benznidazole. *Tc*AKR has been extensively studied, being most efforts focused on characterizing its implication in trypanocidal drug metabolism; however, little is known regarding its biological role. Here, we found that *Tc*AKR is confined, throughout the entire life cycle, into the parasite mitochondria providing new insights into its biological function. In particular, in epimastigotes, *Tc*AKR is associated with the kinetoplast, which suggests additional roles of the protein. The upregulation of *Tc*AKR, which does not affect *Tc*OYE expression, was correlated with an increase in PGF_2_α, suggesting that this enzyme is related to PGF_2_α synthesis in *T. cruzi*. Structural analysis showed that *Tc*AKR contains a catalytic tetrad conserved in the AKR superfamily. Finally, we found that *Tc*AKR is also involved in Nfx metabolization.

## 1. Introduction

Chagas disease, or American Trypanosomiasis, is a chronic parasitosis caused by the protozoan parasite *Trypanosoma cruzi*. It is estimated that nearly 7 million people are infected worldwide [1]. However, this is probably an underestimation since no in-depth studies cover all of Latin America, the endemic area of this disease. In nature, it can be transmitted either through a triatomine vector (vector-dependent transmission that includes ingestion) or transplacentally (vector-independent transmission); in addition, human activities such as blood transfusion, organ transplantation, and laboratory accidents constitute additional sources of infection. It causes high morbidity and mortality rates in endemic and non-endemic areas, leading to the global spread of Chagas disease [2,3,4].

Benznidazole (Bzn) and Nifurtimox (Nfx) are currently the only drugs proven effective against Chagas disease, although they exhibit undesirable side effects that can lead to treatment interruption [5,6]. Both act as prodrugs and must be activated by reducing the nitro-group [7]. In *T. cruzi*, three enzymes have been related to Bzn and Nfx metabolism: the NADH-dependent *Trypanosoma*l type I nitroreductase (*Tc*NTRI) [8], the NADPH-dependent Old Yellow Enzyme (*Tc*OYE) [9,10,11,12] and an aldo-keto reductase name *Tc*AKR [13,14,15,16]. *Tc*AKR belongs to a conserved superfamily present in all domains of life whose function is the reduction of aldehydes and ketones [17,18]. In *Trypanosoma*tids, a member of this family exhibits PGF_2_α synthase activity both in *Trypanosoma brucei* and *Leishmania* spp. [19,20,21], but its putative ortholog in *T. cruzi* did not show such activity [13]. Instead, *Tc*AKR has NADPH-dependent reductive activity on quinones, Bzn, methylglyoxal, and isoprostanes [13,14,16]. However, when the available information is analyzed, some contradictory results appear. On the one hand, this enzyme has been captured through an untargeted chemical proteomics approach using immobilized Bzn, which strongly suggests that it could be related to reductive drug activation and resistance mechanisms [15]. In line with this finding, it has been shown that both recombinant and native *Tc*AKR enzymes reduce Bzn, and *Tc*AKR-overexpressing epimastigotes showed higher NADPH-dependent Bzn reductase activity [14]. However, more recently, it was proposed that the enzyme does not use Bzn as a substrate, and this was studied kinetically and by mass spectrometry [16]. The second controversial point is that most characterized prostaglandin F_2_α synthases (PGFS) belong to the aldo-keto reductase protein family [22], including the PGF_2_α synthases identified in *T. brucei* and *Leishmania* spp. [19,20]. In this context, it is surprising that attempts to demonstrate this activity in *Tc*AKR failed [13,16], taking into account their homologies and the ability of *T. cruzi* to convert exogenous arachidonic acid (AA) into PGF_2_α [23]. A possible explanation is that the presence of an “Old Yellow Enzyme” (*Tc*OYE) in this parasite is entirely responsible for this activity. However, it does not seem like a plausible idea, especially considering that *Tc*OYE was likely acquired more recently through horizontal transfer [11]. To provide clarity in this regard, we performed functional, structural, and phylogenetic studies to obtain new insights into the role of this protein.

## 2. Materials and Methods

### 2.1. Plasmids Construction, Recombinant Protein, and Antibody

*Tc*AKR coding sequence was PCR-amplified from genomic DNA of *T. cruzi* Dm28c epimastigotes with Pfu DNA polymerase (Fermentas, Vilnius, Lithuania) using Fw_TcAKR_XbaI_Dm28c (AATCTAGAATGAAATGCAATTACAGCTGTGTG) and Rv_TcAKR_HindIII_cSTOP_Dm28c (AAAAGCTTTCACTCCTCTCCACCAGAGAAAAAATTATC) primers. PCR products were cloned in pGEM^®^-T Easy plasmid (Promega, WI, USA) using T4 DNA Ligase (Sigma-Aldrich, WI, USA) and sequenced. The coding sequence was subcloned into the pTREX-n vector [24] and the pQE30 vector (Qiagen, Venlo, The Neterlands). Recombinant *Tc*AKR purification under non-native conditions was performed with nickel-charged affinity resin (Ni-NTA). The recombinant protein purity was analyzed by 12% SDS-PAGE stained with colloidal coomassie (Brilliant Blue G-250, Sigma-Aldrich, WI, USA), and protein concentration was determined by the Bradford method [25]. A polyclonal antiserum against *Tc*AKR was obtained from New Zealand White rabbits after intraperitoneal injection of 100 μg of recombinant *Tc*AKR in Freund’s Complete Adjuvant (Sigma-Aldrich, WI, USA), followed by two immunizations with 50 μg of recombinant protein in Freund’s Incomplete Adjuvant (Sigma-Aldrich, WI, USA). Serum was obtained after 15 days of the last boost.

### 2.2. Parasites and Cells

Vero cells [26] were cultivated in Dulbecco’s Modified Eagle’s Medium supplemented with 10% (*v*/*v*) fetal bovine serum (FBS), penicillin (100 U mL^−1^) and streptomycin (100 μg mL^−1^) at 37 °C in a humidified 5% CO_2_ atmosphere. *T. cruzi* Dm28c [27] epimastigotes were cultured axenically in liver infusion tryptose (LIT) medium supplemented with 10% (v v^−1^) inactivated FBS at 28 °C. Trypomastigotes were collected from the supernatant of infected monolayers of Vero cell lines and were maintained cyclically. For stable transfections, *T. cruzi* epimastigotes were transfected with pTREX-n (empty vector) or pTREX-n *Tc*AKR construction. Parasites (8 × 10^7^) were washed three times with PBS, resuspended in HBS Buffer (21 mM HEPES, 137 mM NaCl, 5 mM KCl, 6 mM glucose, pH 7.4), and electroporated with 100 μg of plasmid DNA using two pulses at 450 V, 1300 μF and 13 Ω in 4 mm cuvettes. Transfected parasites were then selected with increasing G418 (Sigma-Aldrich, WI, USA) concentrations from 50 μg mL^−1^ to 300 μg mL^−1^. Overexpression of *Tc*AKR was confirmed by Western blot analysis and indirect immunofluorescence.

### 2.3. PGF_2_α Synthase Activity

PGF_2_α determination was performed with the PGF2 alpha High Sensitivity ELISA Kit (Abcam, Boston, MA, USA). Briefly, *Tc*AKR-overexpressing parasites were washed twice with PBS and incubated with 50 μM of arachidonic acid (Abcam, Boston, USA) for two hours. Cells were removed by pelleting at 1100× *g* for 10 min. Parasite-free supernatant was stored for PGF_2_α measurement, the parasites were resuspended in PBS, and an extract was performed by thermal shock (15 min at −80 °C and 15 min at 37 °C three times consecutively). Finally, the resulting homogenized was centrifuged at 20,000× *g* for 30 min at 4 °C, and the supernatant was used for PGF_2_α determination. The measurements were performed in biological replicates.

### 2.4. Structural Comparison of AKR Proteins

The structural-based phylogenetic analysis was performed with the Multiseq [28] plugin of VMD [29] considering AKRs from *T. cruzi* (*Tc*AKR, PDB: 4GIE), *T. brucei* Homo sapiens (AKR5A2, PDB: 1VBJ), *L. major* (AKR5A, PDB: 4G5D), *Bacillus subtilis* (AKR5G1, PDB: 3D3F), *Rattus norvegicus* (AKR1B14, PDB: 3QKZ) *Homo sapiens* (AKR1C3, PDB: 2F38 and AKR1B1, PDB: 1US0). Three-dimensional structures were first aligned using the algorithm STAMP (Structural Alignment of Multiple Proteins) [30], which minimizes the Cα distance between residues of each molecule by applying globally optimal rigid-body rotations and translations. Then a QH [4] score was computed for each pair of aligned structures to infer the structural similarity (distance) from which trees were plotted:
$$Q_{H}=N\left[q_{aln}+q_{gap}\right]$$
where q_aln_ and q_gap_ are score functions accounting for structurally aligned regions and structural deviations induced by insertions, respectively, and N is a normalization constant.

Electrostatic potentials were calculated with APBS [31] using a cubic grid of 120 Å per side with 10 points per Å2 setting a dielectric constant of 78.54 for water, a salt bath of 0.150 mM NaCl, and a temperature of 298 ºK. The potential was forced to converge to zero at boundaries. PQR file preparation was performed with the PDB2PQR server [32]. AMBER charges were used, and protonation states at pH 7.0 were set according to PROPKA [33]. The dimerization interface was explored with PDBePISA (Proteins, Interfaces, Structures, and Assemblies) [34].

### 2.5. Transcript Abundance Analysis

Transcriptomic analysis of both copies of *Tc*AKR was performed using RNA-seq data from different stages of *T. cruzi* Dm28c [35]. Salmon [36] was used to estimate transcript levels. The normalized counts of *Tc*AKR were plotted using R.

### 2.6. Epimastigotes Synchronization

Cell cycle synchronization was performed as described previously [37,38,39]. Wild-type Dm28c epimastigotes were washed twice with PBS and subsequently incubated with LIT supplemented with 20 mM of hydroxyurea (Sigma-Aldrich, WI, USA) for 24 h. Parasites were washed twice with PBS and cultured in LIT medium supplemented with 10% (v v^−1^) inactivated fetal bovine serum. At different times cells were washed with PBS and fixed for 30 min with 4% (w v^−1^) paraformaldehyde for immunolocalization studies.

### 2.7. Immunolocalization Studies

For indirect immunofluorescence (IIF) localization, parasites were fixed for 16 h at 4 °C with 4% (w v^−1^) paraformaldehyde and then incubated with 50 mM ammonium chloride (Sigma-Aldrich, WI, USA) for 10 min at room temperature. Parasites (1 × 10^6^) were settled in polylysine pre-treated slides and permeabilized for 5 min with 0.5% (v v^−1^) Triton™ X-100 (Sigma-Aldrich, WI, USA). Blocking was performed with 2% (w v^−1^) BSA, 0.1% (v v^−1^) Tween 20 in PBS for 1 h, and washing with 0.1% (v v^−1^) Tween 20 in PBS. Cells were incubated with polyclonal antibodies anti-*Tc*AKR (1/30 dilution) and anti-*Tc*mTXNPx (1/100 dilution) primary antibodies for 2 h. After three washes, Alexa Fluor^®^ 488 goat anti-rabbit IgG) or Cy3^®^ goat anti-mouse IgG (Invitrogen, MA, USA) secondary antibodies were added for 1 h at a 1/1000 dilution. After four washes, slides were mounted with Fluoroshield^TM^ with DAPI (Sigma-Aldrich, WI, USA) and visualized under Zeiss confocal microscope. The whole procedure was performed at room temperature.

Ultrastructure expansion microscopy (UExM) was performed as described previously [40,41] without modifications. All images were acquired using a Zeiss confocal LSM880 microscope using a Plan-Apochromat 63×/1.40 oil objective. All images were acquired and processed using the Zeiss ZEN blue edition v2.0 software. All images were deconvolved using Huygens Professional v19.10.0p2 64b (https://svi.nl/, accessed on 30 December 2022). Alexa Fluor™ 488 NHS Ester (Succinimidyl Ester) (Thermo Fisher Scientific, MA, USA) was used to label the amines (R-NH2) of proteins.

### 2.8. Drug Susceptibility

Susceptibility experiments were performed as previously described [11]. Briefly, *Tc*AKR-overexpressing epimastigotes (5 × 10^6^) were washed twice with 1% (w v^−1^) glucose in PBS and then incubated for 24 h with Nfx (20, 50 and 100 μM) in the same media. Parasites transfected with the empty vector were used as controls. The viability was evaluated with the resazurin reagent (Sigma-Aldrich, WI, USA), measuring absorbance at 490 and 595 nm. Results are referred to the condition of parasites without treatment.

### 2.9. IC_50_ Determination

For IC50 experiments, epimastigotes were seeded at 3 × 10^6^ cells per mL and incubated with serial dilutions of nifurtimox starting from 100 μM. Control conditions of parasites without the drug (100% survival) and culture medium without parasites were included. After 72 h at 28 °C, parasite survival was determined by the resazurin method as described [42]. Results are expressed as the mean of three different and independent experiments and refer to the condition of parasites without treatment.

### 2.10. Phylogenetic Analysis

A representative set of AKR sequences was obtained from NCBI [43]. Sequences with the characteristic domains of the protein family (“Aldo ket red” in Pfam y “Aldo ket red superfamily” in NCBI-CD search) were used to create an AKR database to search these proteins in complete nucleotide sequences of *Trypanosoma*tids. To identify the sequences regardless of genome annotation, the AKR sequences were retrieved from protozoan genomes using the command-line Basic Local Alignment and Search Tool (BLAST)x [44] and several in-house scripts in R. The protein sequences were multiply aligned using the accurate mode of T-coffee [45]. This method considers the 3D structures available for improving the alignment quality. ProtTest v3.2.2 [46] was applied to find an optimal substitution model for each alignment. WAG (Whelan and Goldman) was the best-fit model. Phylogenetic analyses were performed with the maximum-likelihood method using the program PhyML [47]. For the visualization, FigTree v1.4.2 was used.

### 2.11. Statistical Analysis

GraphPad Prism^®^ Version 5.0 was used to determine statistically significant differences. Data are presented as mean ± SE. Statistical significance was assumed with probability values less than or equal to 0.05 using the following convention: * *p* ≤ 0.05; ** *p* ≤ 0.01; *** *p* ≤ 0.001.

## 3. Results

### 3.1. Phylogenetic Analysis of Aldo-Keto Reductase Protein Family in Trypanosomatids

Given that most currently characterized PGFS belong to the AKR superfamily [22] and AKR proteins are taxonomically widely distributed [18], we performed the phylogenetic characterization of AKR proteins from *Trypanosoma*tids. A representative set of well-known AKR (n = 70) sequences was obtained from NCBI (Appendix A). Sequences with the characteristic domains of the family (“Aldo ket red” PF00248 in Pfam and “Aldo ket red superfamily” cl00470 in NCBI-CD search) were used to create an AKR database to search these proteins in 37 genomes, and 161 recovered sequences (Appendix A) were used for phylogenetic reconstructions (Appendix A and Appendix A). The resulting tree shows that the proteins are grouped into four distinct phylogenetic groups (Figure 1). These clades consist of sequences with the same gene annotation. In particular, *Tc*AKR clustered with AKRs annotated as PGFS. However, this clade is subdivided into two groups: one comprising proteins whose PGFS activities have been demonstrated: *T. brucei* (GenBank AB034727.1; seq73, seq75 and seq 79 in our analysis), *L. major* (GenBank J04483.1; seq 42, seq45, and seq52 in our analysis), *L. donovani* (GenBank BAC07250.1, seq106 and seq107 in our analysis), and *L. tropica* (GenBank ABI17868.1; seq110 in our analysis) [19,20]; and another, where *Tc*AKR is found, which includes proteins annotated as PGFS but differ in sequence.

### 3.2. TcAKR Is Located in the Mitochondria and Is Expressed in Epimastigotes and Amastigotes

The Dm28c genome [48] contains two copies of *tcakr* (C4B63_207g13 and C4B63_207g15) encoded in the same chromosome and separated by a gene coding for the 40S ribosomal protein S2 (Figure 2a). *Tc*AKR coding sequence was cloned and expressed in bacterial expression vectors, and the protein was purified from *E. coli* to generate polyclonal antibodies. To address the subcellular localization of *Tc*AKR, we performed indirect immunofluorescence (IIF) assays in wild-type epimastigotes, intracellular amastigotes, and cell-derived trypomastigotes. We found that *Tc*AKR is localized inside the single ramified mitochondria spread throughout the cell body in intracellular amastigotes and co-localizing with the mitochondrial tryparedoxin peroxidase (*Tc*mTXNPx) [49]. Nevertheless, *Tc*AKR was mainly localized at both ends of the kinetoplast in epimastigotes (Figure 2b). In trypomastigotes, the non-replicative stage, the levels of *Tc*AKR were undetectable. The analysis of mRNA expression from RNA-seq data yielded the same differential expression pattern (Figure 2c). By performing ultrastructure expansion microscopy (UExM), we determined that *Tc*AKR has a discrete location in the kinetoplast region proximal to the basal bodies (Figure 2d).

To better understand the particular subcellular localization of *Tc*AKR in epimastigotes, we synchronized their cell cycle using hydroxyurea treatment (HU). To identify specific morphological changes that occur during the *T. cruzi* cell cycle, we used DAPI as a DNA marker for the nucleus (N) and kinetoplast (K) and phase contrast microscopy to assess the number and appearance of flagella (F). As previously described [37,38], G1/S stages are considered between 3 and 6 h after hydroxyurea removal. At these stages (1N1K1F), *Tc*AKR was restricted to the kinetoplast ends (Figure 3). At the onset of the G2 phase, the stage defined when a second flagellum emerges from the cellular pocket (1N1K2F), the signal is more intense at one of the sites where *Tc*AKR is localized. In addition, at this stage, *Tc*AKR was observed in both extremes of the kinetoplast and more distributed in the kinetoplast. In late G2 and during the transition to M, *Tc*AKR is localized in three positions on an elongated kinetoplast. After duplication of the kinetoplast (1N2K2F) and during mitosis (2N2K2F), *Tc*AKR was again located at both extremes of the kinetoplast. These results indicate a dynamic and site-specific localization of *Tc*AKR that changes concomitantly with the progress of the parasite cell cycle.

### 3.3. TcAKR Is Related to PGF_2_α Synthase Activity

To gain insight into the PGF_2_α synthase activity of *Tc*AKR in parasites, we overexpressed the enzyme and measured PGF_2_α production. Western blot analysis showed an increased protein level in *Tc*AKR-overexpressing parasites compared to control parasites transfected with an empty plasmid (Figure 4a). *Tc*AKR overexpression did not interfere with *Tc*OYE expression, as can be observed by cross overexpression controls performed by Western blot. As stated above, PGF_2_α synthase activity could not be previously demonstrated for *Tc*AKR [13]. We first determined PGF_2_α synthesis in epimastigotes, finding an increase in the PGF_2_α production in TcAKR-overexpressing parasites (Figure 4b). To have adequate negative control, we used trypomastigotes, which under normal conditions do not express the enzyme, and TcAKR-overexpressing trypomastigotes. PGF_2_α was measured in the pellets and supernatants of parasites previously incubated with arachidonic acid as substrate, finding an increase in PGF_2_α synthase activity in the overexpressing trypomastigotes, which is statistically significant (Figure 4c).

### 3.4. Structural Analysis of TcAKR

We performed structural comparative analyses of *Tc*AKR protein with AKR structures from *T. brucei*, *L. major*, *Bacillus subtilis*, *Rattus norvegicus,* and *Homo sapiens* to study whether *Tc*AKR has structural similarities to other AKR with PGFS activity. Electrostatic potentials were calculated for every structure to perform a detailed comparative analysis at the active site, and three-dimensional structures were aligned. All catalytic and cofactor binding residues were conserved in all the compared AKR structures (Figure 5). Our results indicate that *Lm*AKR, *Tb*AKR, and *Tc*AKR are homologous proteins with similar protein folding and active site architecture. The comparison with human PGFS highlights preserving the catalytic function through evolution. Our analysis concludes that no structural reason could prevent *Tc*AKR from performing as a PGFS.

In addition, by modeling on the *Tc*AKR crystal structure we cannot neither confirm nor discard a dimer conformation [16]. Analyzing the crystallographic structure of *Tc*AKR NADP-bound (PDB ID: 4FZI) [50] representing the conformation of the enzyme, we show a contact surface where the monomers could interact (Figure 6a). On the other hand, when we increased the incubation time of the primary antibody (16 h), we observed, in addition to the expected band (32 kDa), bands of higher molecular weight (Figure 6b).

### 3.5. TcAKR-Overexpressing Parasites Increases Their Susceptibility to Nifurtimox

*Tc*AKR has been studied in relation to the interaction, metabolization, and resistance to Bzn [14,15,16,51,52]. Nevertheless, there is no information regarding *Tc*AKR and Nfx metabolism in the parasites. Here, we evaluated sensitivity to Nfx of *Tc*AKR-overexpressing cells and calculated IC50. Transfected parasites overexpressing *Tc*AKR were more susceptible to Nfx than the control (Figure 7a), showing a statistically significant decrease in all the concentration assays. Notably, Nfx IC50 in transfectant parasites decreases by nearly four times.

## 4. Discussion

We found that *Tc*AKR is differentially regulated along the *T. cruzi* life cycle. It is expressed in the replicative parasite forms (epimastigotes and amastigotes) at the protein and mRNA levels but is undetectable in trypomastigotes. This expression pattern is the same as that of *Tc*OYE, which turns off entirely in the trypomastigote stage [11]. *Tc*AKR has a particular subcellular localization in the parasite. Through immunofluorescent and UExM assays, we observed that *Tc*AKR is located in the mitochondria. This does not agree with previous reports that located the enzyme in the cytosol and reported its expression in trypomastigotes by Western blot [13]. The latter observation may be related to amastigotes contamination in the preparation, although strain differences cannot be ruled out. The enzyme exhibits a broad distribution in the mitochondria in amastigotes; nevertheless, in epimastigotes, it is located in proximity to the kinetoplast. *Tc*AKR was observed to be located at both ends of the kinetoplast, close to the antipodal sites, two structure assemblies where ligases, topoisomerases, polymerases, and other proteins are located [53,54] and where the repair of gaps in the newly synthesized minicircles that are reattached to the kDNA network occurs [55,56]. Since the kDNA replication is closely related to the cell cycle, we synchronized epimastigotes and followed *Tc*AKR along the process, finding that its location is linked to the changes in the kinetoplast during S, G1, G2, and mitosis and cytokinesis. It was previously reported that methylglyoxal is the most efficient substrate for *Tc*AKR compared with other toxic ketoaldehydes [16]. The primary endogenous sources of methylglyoxal are glycolysis and the breakdown of threonine and lipid peroxidation [57]. These last reactions take place in mitochondria, in the exact subcellular location that *Tc*AKR, indicating that this enzyme could have a relevant role in detoxification at the mitochondrial level, especially protecting kDNA replication, since methylglyoxal severely inhibits DNA replication [58]. Finally, high-resolution microscopy by UExM reveals that TcAKR is located in the kinetoplast region proximal to basal bodies. These localization changes along the life cycle deserve future studies using high-resolution microscopy methods.

By phylogenetic studies, we found that the AKR family in trypanosomatids are grouped in four main clades and TcAKR clusters with those with PGFS function in *Leishmania* and *T. brucei* [19,20,21]. However, in T. cruzi, this activity appears to be missing [13,16]. The absence of *Tc*AKR expression in trypomastigotes constitutes an opportunity to revisit PGFS activity using *Tc*AKR-overexpressing parasites. We observed that the level of PGF_2_α measured in transfectant parasites was significantly higher than in control parasites. Our experimental model allows us to conclude that TcAKR can catalyze the synthesis of PGF2α. In concordance with previous reports [11], high levels of PGF_2_α were found in measurements made from the culture supernatant of *Tc*AKR-overexpressing parasites, suggesting that this molecule is released from the parasite once it is synthesized. In this sense, it was also reported that secretion of PGF_2_α synthesized by *T. brucei* and *L. infantum* [19,59]. Ashton and collaborators demonstrated, using TXA_2_ synthase-null mice, that most of the TXA_2_ in infected mice is parasite-derived [23]. All these studies point out that parasite-derived eicosanoids exert a paracrine or endocrine effect in *Trypanosoma*tid infections. We determined that the catalytic tetrad, highly conserved throughout the AKR superfamily [60], is present in *Tc*AKR, with a core folding of (α/β) TIM barrel that acts via classic ping-pong kinetics [17,61]. The phylogenetic tree shows that *Tc*AKR groups with PFGS; however, this clade is subdivided into two groups: one comprising proteins whose PGFS activities have been demonstrated and another where *Tc*AKR is found, which includes proteins annotated as PGFS but with some differences in sequences.

There is a discrepancy in the literature regarding the evidence of the oligomerization state of *Tc*AKR [13,16]. Garavaglia et al. (2016) detected mixed species containing monomeric, dimeric, and tetrameric forms of the enzyme from the CL-Brener strain by gel chromatography with the recombinant soluble enzyme [13], while Roberts et al. (2018) found a single peak with a calculated MW close to that of a monomer [16]. When we increased the incubation time in western blots, bands compatible with TcAKR oligomers were detected, although mass spectrometry must confirmed their identity. Also, we found a contact surface where the monomers could interact, but we cannot discard that as a consequence of the crystallization process.

*Tc*AKR has been widely studied in relation to drug metabolism, including Bzn. Its overexpression leads to increased resistance to Bzn [14]. However, recent studies have not found evidence for the reduction of Bzn by recombinant *Tc*AKR and homologs in the related parasites *T. brucei* and *L. donovani*. Instead, these enzymes metabolize a variety of toxic ketoaldehydes, including glyoxal and methylglyoxal, suggesting a role in cellular defense against chemical stress [16]. Bzn is a prodrug and undergoes activation via two sequential two-electron reduction steps by a mitochondrial nitroreductase to form a toxic reactive hydroxylamine intermediate. This compound subsequently undergoes rearrangement to a dihydro-dihydroxy intermediate [62] or forms adducts with low molecular mass thiols [51]. This intermediate can then dissociate to form glyoxal and N-benzyl-2-guanidinoacetamide. UPLC-QToF/MS analysis of Bzn bioactivation by *T. cruzi* cell lysates confirmed previous reports identifying numerous drug metabolites, including a dihydro-dihydroxy intermediate that can separate into N-benzyl-2-guanidinoacetamide and glyoxal. However, metabolomic studies did not detect glyoxal adducts [51]. *Tc*AKR may play a role in detoxifying some of the toxic metabolites generated by the reductive metabolism of Bzn by NTRI in the mitochondria, including glyoxal or others; also considering *Tc*AKR was found in previous reports bound to immobilized Bzn [15]. In this work, we focused our study on Nfx, that, as most nitroheterocyclic agents, must undergo activation by nitroreduction, a process catalyzed by type I nitroreductases. It has been shown that these reductases generate cytotoxic nitrile metabolites responsible for trypanocidal activity [63]. We found that *Tc*AKR-overexpressing parasites are more susceptible to Nfx, which constitutes a difference from *Tb*AKR that does not affect sensibility to Nfx in *T. brucei*. Metabolomic and proteomic studies on Nfx metabolization are necessary to evaluate the direct or indirect (by favoring the overexpression of other proteins) role of this enzyme in the metabolism of this drug, where *Tc*AKR overexpression parasites constitute a valuable tool that can provide clues on the unknown mode of action of Nfx.

In summary, the substrate-specificity of this aldo-keto reductases is broader than other prostaglandin F_2_α synthases, and *Tc*AKR is likely to be involved in arachidonic metabolism, detoxification of ketoaldehydes and the activation of the prodrug nifurtimox. Its subcellular localization during the cell cycle is compatible with a significant role in kDNA metabolism by eliminating toxic metabolites that can inhibit replication. Together these results position *Tc*AKR as a relevant target for the action of drugs.

## Figures and Tables

**Figure 1 pathogens-12-00085-f001:**
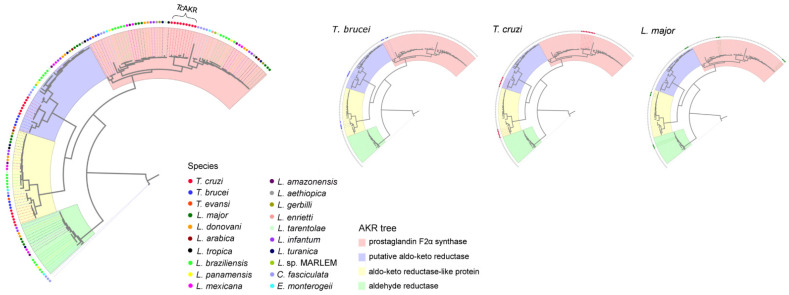
Evolutionary relationship of Aldo-Keto Reductase proteins in *Trypanosoma*tids. Evolutionary relationships of AKR sequences retrieved from protozoan genomes. The sequence alignment was performed using the accurate mode of T-Coffee software. The tree was built with the maximum-Likelihood method. The sequences are provided in Appendix A.

**Figure 2 pathogens-12-00085-f002:**
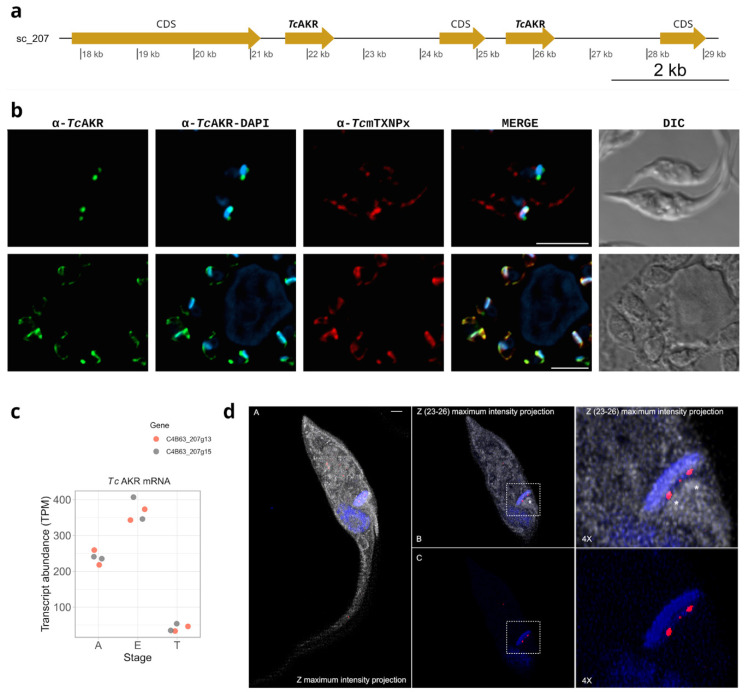
*Tc*AKR subcellular localization. (**a**) Genomic localization of *tcakr* genes in PRFA01000207 scaffold. (**b**) Immunolocalization of *Tc*AKR and *Tc*mTXNPx in epimastigotes and amastigotes using rabbit α-*Tc*AKR (1/30) (green) and α-*Tc*mTXNPx (1/100) (red) polyclonal antibodies. DAPI was used as a nucleus and kinetoplast marker. Scale bar: 5 µm. (**c**) RNA-seq analysis to quantify the mRNA levels of the *Tc*AKR during the life stages: A = amastigote, E = epimastigote, and T = trypomastigote. (**d**) Epimastigotes were subjected to ultrastructure expansion microscopy (UExM) and stained with rabbit α-*Tc*AKR (1/30) (red), N-Hydroxysuccinimide (NHS) (gray), and DAPI. Note that NHS labels the pan-proteome, labeling different parasite structures as the nucleus, flagella, and kDNA, among others. (A) Z Maximum intensity projection showing *Tc*AKR (red), NHS (gray), and kDNA and nucleus with DAPI (blue). (B) Z (23–26) Maximum intensity projection showing *Tc*AKR (Red), NHS (gray) and kDNA and nucleus with DAPI (blue). (C) Z (23–26) Maximum intensity projection showing *Tc*AKR (red) and kDNA and nucleus with DAPI (blue). Scale bar: 1 µm. BB: basal bodies.

**Figure 3 pathogens-12-00085-f003:**
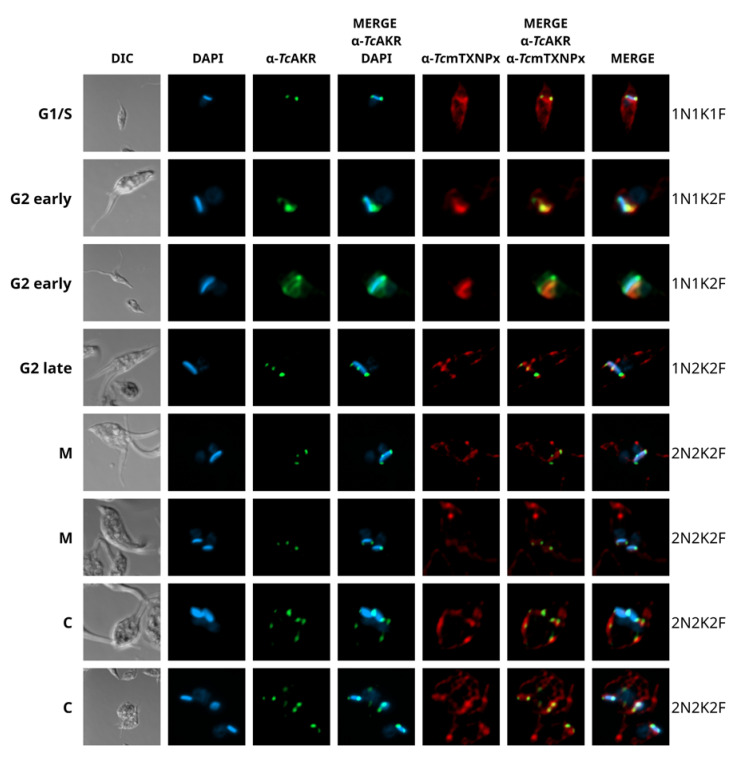
*Tc*AKR localization during the epimastigotes cell cycle. Localization of *Tc*AKR by immunofluorescence in synchronized epimastigotes stained with α-*Tc*AKR (1/30) (green) and α-*Tc*mTXNPx (1/100) (red) rabbit polyclonal antibodies. Cell cycle stages (G1/S, G2, M, and C) are defined by the position of the DAPI-stained kinetoplast (K), nucleus (N) in blue, and the number of flagella (F) analyzed in DIC.

**Figure 4 pathogens-12-00085-f004:**
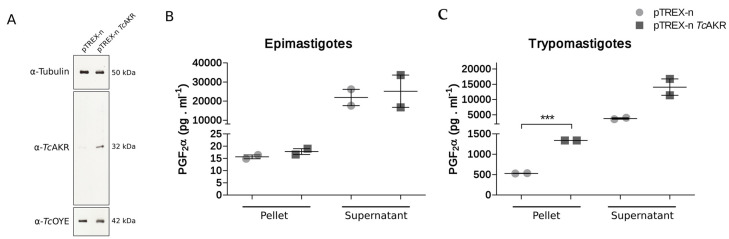
TcAKR overexpression and PGF_2_α detection. (**A**) Western blot analysis of total protein extracts of trypomastigotes transfected with pTREX-n (empty vector) and pTREX-n *Tc*AKR (includes complete *Tc*AKR coding sequence) using rabbit α-*Tc*AKR polyclonal antiserum. The relative expression was estimated by image densitometry analysis normalized by tubulin expression. *Tc*OYE expression was evaluated in *Tc*AKR-overexpressing parasites using rabbit α-*Tc*OYE polyclonal antiserum as a control. (**B**) Determination of PGF_2_α in cells and supernatant of *Tc*AKR-overexpressing epimastigotes. (**C**) Determination of PGF_2_α in cells and supernatant of *Tc*AKR-overexpressing cell-derived trypomastigotes. These results correspond to two biological replicates. Asterisks represent statistical significance (two-tailed Student’s *t*-Test) *p*-values: pellet *p* = 0.0001, supernatant *p* = 0.0623.

**Figure 5 pathogens-12-00085-f005:**
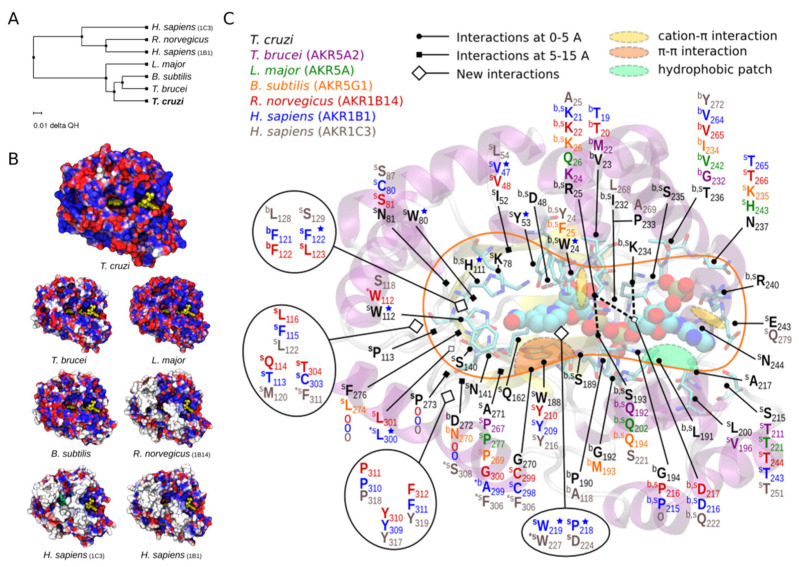
Structural comparatives analyses of *Tc*AKR protein with AKR structures from *T. brucei*, *L. major*, *B. subtilis*, *R. norvegicus,* and *H. sapiens*. (**A**) Phylogenetic tree of AKRs based on the QH index. (**B**) Electrostatic potential mapped on solvent-accessible surfaces of AKR proteins. Values range from −130 (red) to 130 mV (blue). (**C**) Structural alignment and sequence conservation of AKR proteins. The first and second shells of residues around the active site of *Tc*AKR. Residues that differ from other species are indicated. The structures used were: *T. cruzi Tc*AKR PDB = 4GIE; *T. brucei* AKR5A2 PDB = 1VBJ; *L. major* AKR5A PDB = 4G5D; *B. subtilis* AKR5G1 PDB = 3D3F; *R. norvegicus* AKR1B14 PDB = 3QKZ; *H. sapiens* AKR1C3 PDB = 2F38.

**Figure 6 pathogens-12-00085-f006:**
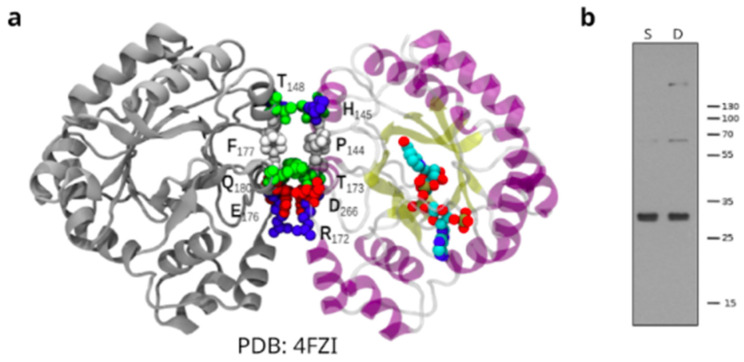
*Tc*AKR studies on oligomerization. (**a**) Interaction surface of the proposed monomers using *Tc*AKR crystals. PDB 4FZI: interface A/B x, y, z. Area: 377.7 A^2. DG: −5.5 Kcal/mol. DG *p*-value = 0.21. (**b**) Western blot analysis of total protein extracts from Sylvio and Dm28c epimastigotes using rabbit α-*Tc*AKR polyclonal antiserum overnight. Protein bands corresponding to molecular weights compatible with oligomerization states are observed. S: Sylvio, D: Dm28c.

**Figure 7 pathogens-12-00085-f007:**
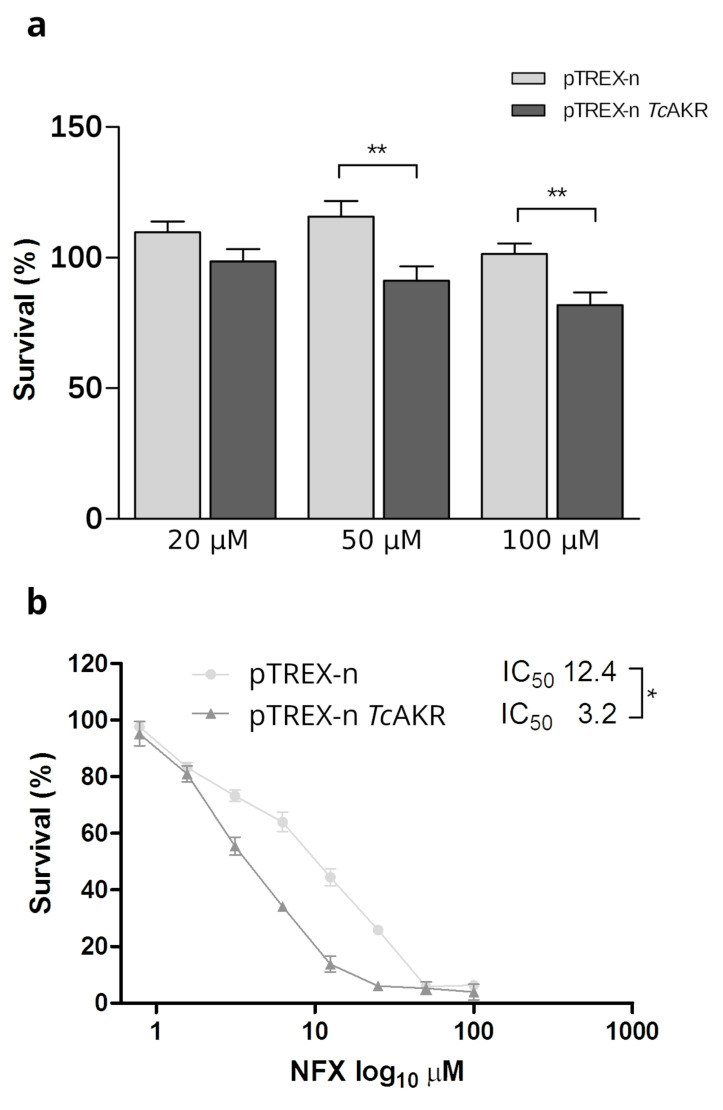
Susceptibility to Nfx. (**a**) Viability percentages of transfected epimastigotes challenged with different concentrations of Nfx. The percentages of cell viability are normalized against parasites without treatment. Values are the means of three independent assays performed in quadruplicate. Nfx *p*-values: 20 μM *p* = 0.0868; 50 μM *p* = 0.0062; 100 μM *p* = 0.0054. (**b**) Percentage of epimastigotes survival. IC50 for NFX was determined in pTREX and pTREXn-TcAKR parasites. Values are the means of three independent assays performed in quadruplicate. (*p* = 0.0227; asterisks represent statistical significance; two-tailed Student’s *t*-test). (**a**) and (**b**) are two independent experiments (different datasets).

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
