# Peer review of "New Insights into the Role of the Trypanosoma cruzi Aldo-Keto Reductase TcAKR"

_pathogens, 2023, doi:10.3390/pathogens12010085_

Round 1

Reviewer 1 Report

The paper by Diaz-Viraque et al. addresses the role of aldo-keto reductase TcAKT in Trypanosoma cruzi. This protein has been previously characterized and identified as one of enzymes involved in the activation of anti-trypanosome drugs benznidazole and nifurtimox, but there are some controversies related to the expression, localization and enzymatic activity that the authors of this manuscript seek to solve.

The authors found that TcAKT is encoded by two genes in T. cruzi, and their phylogenetic analysis groups the genes with those of PGFs in trypanosomatids. Localization experiments indicate a variable localization depending on the life stages, with some evidence of accumulation on antipodal sites and/or basal bodies. When TcAKT was overexpressed in trypomastigotes, the authors detected an increase in the production of PGF2, supporting the role of TcAKT in PGF synthesis, in contrast with previous reports. The authors used in silico structural analysis of TcAKT active sites to analyze whether differences in the sequence and 3D structure could explain the functional discrepancies and conclude that there are no differences that could explain the lack of PGF synthetic activity of TcAKT but is not clear how they arrived at this conclusion. Finally, they evaluated whether overexpression of TcAKT affects the susceptibility to Nifurtimox, finding a modest decrease in the EC50. The putative role of TcAKT in the metabolism of currently used drugs to treat Chagas disease is relevant and grants further exploration, but the current manuscript has significant inconsistencies that need to be addressed to evaluate the contribution of these results to above mentioned controversies.

-       First, it is not clear what the localization of TcAKT is.  The authors indicate that is localized in the mitochondria of amastigotes, but the colocalization images do not show a clear correlation with the mitochondrial marker used. Higher resolution images are. required to support this affirmation. In epimastigotes, the authors indicated the localization of TcAKT in the antipodal sites of the kinetoplast and/or the basal bodies. The expansion microscopy images seem consistent with basal body localization but co-localization studies with a basal body marker and high resolution microscopy are required to clarify this point, as both localizations could have drastically different roles. The authors suggest the role of TcAKT in the maintenance of kDNA but no evidence of such is provided.

-       The resolution of the images in Figure 2 needs to be improved.

-       The overexpression of TcAKT and measurement of activity on PFG synthesis was done in trypomastigotes, a life stage that usually doesn’t express TcAKT, as the authors show in Figure 1 C. It is not clear the biological relevance of these experiments, and whether the results will be applicable to amastigotes or epimastigotes. PGF synthesis experiments. should be done in epimastigotes instead and ideally, comparing TcAKT overexpression with knockouts.

-       In Figure 6B, the authors show a western blot of epimastigotes homogenates incubated with anti-TcAKT antibody overnight and show additional bands, claiming that they correspond to dimers. This is not sufficient evidence of oligomerization and could be also explained by increase in background, and/or interaction of TcAKT with other proteins. Further experiments are needed to define this point.

-       Figure 7A is confusing. It seems like control parasites are surviving 100uM nifurtimox, and this contradicts Figure 7B. Are those two figures showing the same data set? Please, revise the figure.

-       What is the conclusion and relevance of Figure 1?

-       Figure legends are not fully explaining the panels. Please, revise.  

There is a number of speculative claims in the discussion that are not supported by the literature or the results presented in this manuscript (TcAKT mitochondrial localization, formation of dimers, role in kDNA maintenance, detoxification role linked to replication, etc) and should be revised.

Author Response

Reviewer 1

The paper by Diaz-Viraque et al. addresses the role of aldo-keto reductase TcAKT in Trypanosoma cruzi. This protein has been previously characterized and identified as one of enzymes involved in the activation of anti-trypanosome drugs benznidazole and nifurtimox, but there are some controversies related to the expression, localization and enzymatic activity that the authors of this manuscript seek to solve.

The authors found that TcAKT is encoded by two genes in T. cruzi, and their phylogenetic analysis groups the genes with those of PGFs in trypanosomatids. Localization experiments indicate a variable localization depending on the life stages, with some evidence of accumulation on antipodal sites and/or basal bodies. When TcAKT was overexpressed in trypomastigotes, the authors detected an increase in the production of PGF2, supporting the role of TcAKT in PGF synthesis, in contrast with previous reports. The authors used in silico structural analysis of TcAKT active sites to analyze whether differences in the sequence and 3D structure could explain the functional discrepancies and conclude that there are no differences that could explain the lack of PGF synthetic activity of TcAKT but is not clear how they arrived at this conclusion. Finally, they evaluated whether overexpression of TcAKT affects the susceptibility to Nifurtimox, finding a modest decrease in the EC50. The putative role of TcAKT in the metabolism of currently used drugs to treat Chagas disease is relevant and grants further exploration, but the current manuscript has significant inconsistencies that need to be addressed to evaluate the contribution of these results to above mentioned controversies.

-       First, it is not clear what the localization of TcAKT is.  The authors indicate that is localized in the mitochondria of amastigotes, but the colocalization images do not show a clear correlation with the mitochondrial marker used. Higher resolution images are. required to support this affirmation. In epimastigotes, the authors indicated the localization of TcAKT in the antipodal sites of the kinetoplast and/or the basal bodies. The expansion microscopy images seem consistent with basal body localization but co-localization studies with a basal body marker and high resolution microscopy are required to clarify this point, as both localizations could have drastically different roles. The authors suggest the role of TcAKT in the maintenance of kDNA but no evidence of such is provided.

-       The resolution of the images in Figure 2 needs to be improved.

We agree with the reviewer, and in our opinion, the location of AKR is in a previously undescribed region, located between the antipodal sites and the basal bodies. For this reason, we have changed the text in the abstract (line 26) and the results (Lines 242; 246-247) and discussion (Lines 382-383; 395-398) sections

Concerning Figure 2, the objective is to show how TcAKR changes its location in the epimastigote stage, where it ceases to have a diffuse distribution in the mitochondria to concentrate in two focal points that do not strictly correspond neither to the basal bodies nor with the antipodal sites. The mitochondria marker (anti-TcmTXNPx antibody, red) is widely validated, and no background is seen. Note that the distribution of TcMPX is the same between amastigotes and epimastigotes, while for TcAKR, there is a change in localization. In addition, I would like to highlight that the Ultra Expansion Microscopy experiment is highly novel. It is the second publication that uses this new technology in Trypanosoma cruzi (the first was while our article was written: DOI 10.1007/s00436-022-07619-z), which gives an added value to the location by a high-resolution technique.

-       The overexpression of TcAKT and measurement of activity on PFG synthesis was done in trypomastigotes, a life stage that usually doesn’t express TcAKT, as the authors show in Figure 1 C. It is not clear the biological relevance of these experiments, and whether the results will be applicable to amastigotes or epimastigotes. PGF synthesis experiments. should be done in epimastigotes instead and ideally, comparing TcAKT overexpression with knockouts.

We fully understand the reviewer's comment. Precisely because the enzyme is not expressed in trypomastigotes, we understood that this was an opportunity to show whether the enzyme could participate in the metabolism of prostaglandins. The objective was just that; we understand that the model allows us to show it. We were careful with this point, writing: "TcAKR is related to the activity of PGF2α synthase".

In the discussion, we clarify this point in lines 402-403: "The absence of TcAKR expression in trypomastigotes constitutes an opportunity to revisit PGFS activity".

We also added the sentence:

"Our experimental model allows us to conclude that  TcAKR can catalyze the synthesis of PGF2α." (lines 405-406),

-       In Figure 6B, the authors show a western blot of epimastigotes homogenates incubated with anti-TcAKT antibody overnight and show additional bands, claiming that they correspond to dimers. This is not sufficient evidence of oligomerization and could be also explained by increase in background, and/or interaction of TcAKT with other proteins. Further experiments are needed to define this point.

We agree with the reviewer and modified the text (lines 340-342)

-       Figure 7A is confusing. It seems like control parasites are surviving 100uM nifurtimox, and this contradicts Figure 7B. Are those two figures showing the same data set? Please, revise the figure.

For some reason, we did not include the complete text of the figure. We have now completed the legend of Figure 7. We apologize for the mistake.

-       What is the conclusion and relevance of Figure 1?

We thank the reviewer for the question. On the one hand, the relevance is to show the four clades, which can help future studies in this family. And second and more relevant for this study is that TcAKR groups with PGFS. This was the initial reason we decided to explore this activity in the parasite.

We now included in the discussion the two main conclusions of Figure 1 in lines 399-400.

" By phylogenetic studies, we found that the AKR family in trypanosomatids are grouped in four main clades and TcAKR clusters with those with PGFS function in Leishmania and T. brucei [19-21]. However, in T. cruzi, this activity appears to be missing. [13,16].

-       Figure legends are not fully explaining the panels. Please, revise. 

Done.

There is a number of speculative claims in the discussion that are not supported by the literature or the results presented in this manuscript (TcAKT mitochondrial localization, formation of dimers, role in kDNA maintenance, detoxification role linked to replication, etc.) and should be revised.

We have now modified the discussion (highlighted in grey).

Reviewer 2 Report

Trypanosoma cruzi parasites, the causative agents of Chagas disease, are currently only targeted by two drugs. These drugs need to be reduced by parasite enzymes, and there is still some debate as to which parasite enzymes accomplish this enzymatic activity. In the manuscript “New insights into the role of the Trypanosoma cruzi aldo-keto reductase TcAKR”, Florencia Díaz-Viraqué and colleagues try to uncover the enzymatic activity of the protein TcAKR of the parasite, because some reports suggest that this enzyme is a prostaglandin F2alpha synthase, while other suggest that it metabolizes benznidazole (Bzn, one of only two anti-T. cruzi drugs).

Using a combination of in silico tools (sequence and structure alignments, Figures 1, 5 and 6), the authors show that TcAKR is more similar to other prostaglandin F2alpha synthases from Trypanosomatids (family of organisms that contains T. cruzi), than to other aldo-keto reductases. The authors also analyze the expression of TcAKR mRNA and protein and find that this protein is: i) not expressed in trypomastigote; ii) localised to the mitochondria in amastigote; and iii) localised to the basal bodies in epimastigote (Figure 2 and 3). Importantly, the authors show that overexpression of TcAKR leads to an increased synthesis of prostaglandin F2alpha (Figure 4) and increases the susceptibility of the parasite to Nifurtimox (the other effective drug against T. cruzi) (Figure 7).

I think the paper is well structure and scientifically sound, however it does not answer the question it posed, "what is the activity of TcAKR?". The authors show that TcAKR is both a prostaglandin F2alpha synthase and responsible for Nifurtimox susceptibility (which in turn suggests that it might activate nifurtimox). Also, the authors never address the role of TcAKR in benznidazole toxicity.

Moreover, I think it is especially crucial to control for the activity of the NADH-dependent Trypanosomal type I nitroreductase (TcNTRI), as it might explain the increased susceptibility of the over-expressing parasite to nifurtimox.

Without two corrections, I cannot accept the paper:

1) When authors overexpress TcAKR, they control for expression of Old Yellow Enzyme (TcOYE) but they do not control for TcNTRI activity. (For example, like Garavaglia, et al. (2016) Antimicrob Agents Chemother). Without knowing if the level of NTRI activity is the same in the TcAKR-overexpressing parasites, we cannot conclude that the susceptibility to nifurtimox is dependent on TcAKR.

2) Biological and technical replicates – The authors should provide the number of times they repeated each experiment and how many technical replicates were analysed in each. This is especially important for Figures 4 and 7.

Below are also some minor comments.

3) Results Section 3.2 – Authors should reference the dataset/paper they used for the expression of TcAKR through the life cycle.

4) Results Section 3.3 – In which parasite stage was the Western Blot performed? It should be at the trypomastigotes stage, in which the activity assays were performed.

5) Results Section 3.5 – Why do the authors perform two viability experiments with the same compound at different timepoints (Figure 7a and 7b)?

6) If would be interesting to test the susceptibility/IC50 of over-expressing parasites to benznidazole.

7) Figure 1 – authors should highlight where is TcAKR in the tree.

8) Figures 2 and 3 – authors should add scale bar (Figure 3) and make scale bar more visible (Figure 2)

9) Figure 2b – Authors should consider using arrows to identify mitochondria/basal bodies

10) Figure 3 – Brightfield (BF) channel does not exactly match fluorescent channels. Authors should consider cropping/zooming the BF image to match what is observed in fluorescence.

11) Figure 7 – Legend of the Figure should emphasize that viability was measured at different timepoints in a (24h after treatment) and b (72h after treatment).

Author Response

Reviewer 2

Trypanosoma cruzi parasites, the causative agents of Chagas disease, are currently only targeted by two drugs. These drugs need to be reduced by parasite enzymes, and there is still some debate as to which parasite enzymes accomplish this enzymatic activity. In the manuscript “New insights into the role of the Trypanosoma cruzi aldo-keto reductase TcAKR”, Florencia Díaz-Viraqué and colleagues try to uncover the enzymatic activity of the protein TcAKR of the parasite, because some reports suggest that this enzyme is a prostaglandin F2alpha synthase, while other suggest that it metabolizes benznidazole (Bzn, one of only two anti-T. cruzi drugs).

Using a combination of in silico tools (sequence and structure alignments, Figures 1, 5 and 6), the authors show that TcAKR is more similar to other prostaglandin F2alpha synthases from Trypanosomatids (family of organisms that contains T. cruzi), than to other aldo-keto reductases. The authors also analyze the expression of TcAKR mRNA and protein and find that this protein is: i) not expressed in trypomastigote; ii) localised to the mitochondria in amastigote; and iii) localised to the basal bodies in epimastigote (Figure 2 and 3). Importantly, the authors show that overexpression of TcAKR leads to an increased synthesis of prostaglandin F2alpha (Figure 4) and increases the susceptibility of the parasite to Nifurtimox (the other effective drug against T. cruzi) (Figure 7).

I think the paper is well structure and scientifically sound, however it does not answer the question it posed, "what is the activity of TcAKR?". The authors show that TcAKR is both a prostaglandin F2alpha synthase and responsible for Nifurtimox susceptibility (which in turn suggests that it might activate nifurtimox). Also, the authors never address the role of TcAKR in benznidazole toxicity.

Moreover, I think it is especially crucial to control for the activity of the NADH-dependent Trypanosomal type I nitroreductase (TcNTRI), as it might explain the increased susceptibility of the over-expressing parasite to nifurtimox.

We thank the reviewer for these comments.

Without two corrections, I cannot accept the paper:

1) When authors overexpress TcAKR, they control for expression of Old Yellow Enzyme (TcOYE) but they do not control for TcNTRI activity. (For example, like Garavaglia, et al. (2016) Antimicrob Agents Chemother). Without knowing if the level of NTRI activity is the same in the TcAKR-overexpressing parasites, we cannot conclude that the susceptibility to nifurtimox is dependent on TcAKR.

We fully agree with the reviewer's comment. In the original version, we said that TcAKR could have a direct or indirect role, and now we have modified this in the discussion according to the reviewer's suggestion. Regarding NTRI, although it is the best-characterized enzyme in the metabolization of NFX, there are also other recently described proteins (for example, DOI: 10.3389/fcimb.2022.749476 ), and probably there are more due to the pleiotropic effect of Nfx. Therefore, following the reviewer's suggestion, we modified the discussion (lines 451-455).

2) Biological and technical replicates – The authors should provide the number of times they repeated each experiment and how many technical replicates were analysed in each. This is especially important for Figures 4 and 7.

 We thank the reviewer for the suggestion. We now modified the legends of Figures 4 and 7.

Below are also some minor comments.

3) Results Section 3.2 – Authors should reference the dataset/paper they used for the expression of TcAKR through the life cycle.

We apologize for the omission. We now included reference 49 (Pineyro et al., 2008).

4) Results Section 3.3 – In which parasite stage was the Western Blot performed? It should be at the trypomastigotes stage, in which the activity assays were performed.

Yes, it was performed in trypomastigotes. We now included it in the legend of Figure 4.

5) Results Section 3.5 – Why do the authors perform two viability experiments with the same compound at different timepoints (Figure 7a and 7b)?

In 7a we determined viability. In 7b we evaluated susceptibility. This second experiment allowed us to determine IC50. In addition, because of a typing error, we did not include the complete text of the figure. We have now completed the legend of Figure 7. We apologize for the mistake.

6) If would be interesting to test the susceptibility/IC50 of over-expressing parasites to benznidazole.

We have done it, and we did not find differences in IC50. That is why we did not include this result.

7-11) Figures
All changed were done.

Round 2

Reviewer 1 Report

We thank the authors for addressing some of the issues raised on the first revision. Nevertheless, the main experimental concerns were not corrected. In order to support their conclusions, as indicated in the previous review it will be necessary to:

-perform co-localization studies with a basal body marker and high resolution microscopy to establish the localization on a "new" area between the mitochondria and the basal body. 

-Improve the resolution of the images in Figure 2. The colocalization with the mitochondrial marker indicates that TcAKT is in some discrete foci that show some superimposing of the colors, but with that level of resolution is impossible to establish colocalization. The beatiful expansion microscopy images support even further my point. If this is a "new" domain, it needs to be further demonstrated. 

- Measure the activity of TcAKT in epimastigotes to establish its physiological relevance. While the activity by overexpression in trypomastigotes can be informative, the role on PFG synthesis must be confirmed in epimastigotes or epimastigotes to confirm the role suggested by the authors.

- In the new version, you are still indicating that the extra bands on the western blots of Figure 6 are "consistent with oligomers" but no further experimental evidence for this affirmation is provided.

- Figure 7 is still confusing. Are panels A and B the same data set? In A you indicate is "viability percentages" and in B is 'percentage of epimastigotes survival" and in both cases they are treated with increasing amounts of nifurtimox. Are these 2 panels different representation of the same data or not?

Author Response

1- Improve the resolution of the images in Figure 2. The colocalization with the mitochondrial marker indicates that TcAKT is in some discrete foci that show some superimposing of the colors, but with that level of resolution is impossible to establish colocalization. The beatiful expansion microscopy images support even further my point. If this is a "new" domain, it needs to be further demonstrated. 

-perform co-localization studies with a basal body marker and high resolution microscopy to establish the localization on a "new" area between the mitochondria and the basal body. 

We thanks to the reviewer for these considerations. We agree that TcAKR changes in the location need a more in deep study. In our opinion, it should be done in new work (and in collaboration with experts in high-resolution microscopy). We think we must "dissect" the localization changes during the cell cycle, use more markers (e.g., basal bodies), and tomography/three-dimensional animations in future work.

Our proposal to the reviewer is to add the phrase "future studies...". This is because, in this work, we make a global characterization, which includes structural, functional and expression/localization aspects. And it leaves the question to answer in future work on the location changes.

Concerning the suggestions about the figure, we have now modified figure 2 and its legend. First, we changed panel b, including images with more parasites, which gives a more precise overview of the localization of AKR. In the expansion microscopy, we added "*" to show where basal bodies are located.

2- Measure the activity of TcAKT in epimastigotes to establish its physiological relevance. While the activity by overexpression in trypomastigotes can be informative, the role on PFG synthesis must be confirmed in epimastigotes or epimastigotes to confirm the role suggested by the authors.

We have now added the activity in epimastigotes and modified Figure 4 and its legend.

3- In the new version, you are still indicating that the extra bands on the western blots of Figure 6 are "consistent with oligomers" but no further experimental evidence for this affirmation is provided.

We have now modified the text, deleting "consistent with oligomers".

4- Figure 7 is still confusing. Are panels A and B the same data set? In A you indicate is "viability percentages" and in B is 'percentage of epimastigotes survival" and in both cases they are treated with increasing amounts of nifurtimox. Are these 2 panels different representation of the same data or not?

Thanks for the comment. Now at the end of Figure 7 legend we added the sentence: " a) and b) are two independent experiments (different datasets)"

Reviewer 2 Report

After reading the revised version of the manuscript " New insights into the role of the Trypanosoma cruzi aldo-keto reductase TcAKR", from Florencia Díaz-Viraqué and colleagues, I feel the manuscript warrants.

The authors have addressed the issues I raised before: by giving more detail about the replication of the experiment and by explaining the other enzymes that might contribute to the detoxification of Nfx.

I would just add that reference #49 is possibly on the wrong sentence. It should be at the end of the sentence in line 243-245, not in line 240.

Author Response

After reading the revised version of the manuscript " New insights into the role of the Trypanosoma cruzi aldo-keto reductase TcAKR", from Florencia Díaz-Viraqué and colleagues, I feel the manuscript warrants.

The authors have addressed the issues I raised before: by giving more detail about the replication of the experiment and by explaining the other enzymes that might contribute to the detoxification of Nfx.

We thank the reviewer for the comments.

I would just add that reference #49 is possibly on the wrong sentence. It should be at the end of the sentence in line 243-245, not in line 240.

We agree with this suggestion. We have now added the reference at the end of the sentence.